# Zeolitic Imidazolate Framework-8 (ZIF-8) as a Drug Delivery Vehicle for the Transport and Release of Telomerase Inhibitor BIBR 1532

**DOI:** 10.3390/nano13111779

**Published:** 2023-05-31

**Authors:** Shunyu Zhang, Jinxia Li, Liang Yan, Yue You, Feng Zhao, Jixing Cheng, Limin Yang, Yanqi Sun, Qingchao Chang, Ru Liu, Yunhui Li

**Affiliations:** 1Key Laboratory of Environmental Medicine Engineering of Ministry of Education, School of Public Health, Southeast University, Nanjing 210000, China; syzhang@ihep.ac.cn (S.Z.); chengjx0210@163.com (J.C.); 2CAS Key Lab for Biomedical Effects of Nanomaterials and Nanosafety, Institute of High Energy Physics, Chinese Academy of Sciences, Beijing 100049, China; lijinxia@ihep.ac.cn (J.L.); yanliang@ihep.ac.cn (L.Y.); youyue@ihep.ac.cn (Y.Y.); yanglm@ihep.ac.cn (L.Y.); changqc@ihep.ac.cn (Q.C.); liuru@ihep.ac.cn (R.L.); 3Department of Prevention and Health Care, Rizhao 276800, China; tangdou888@163.com

**Keywords:** ZIF-8, drug loading, transport, protonation effect, telomerase inhibitor

## Abstract

Telomerase is constitutively overexpressed in the majority of human cancers and telomerase inhibition provides a promising broad-spectrum anticancer therapeutic strategy. BIBR 1532 is a well-known synthetic telomerase inhibitor that blocks the enzymatic activity of hTERT, the catalytic subunit of telomerase. However, water insolubility of BIBR 1532 leads to low cellular uptake and inadequate delivery and thus, limits its anti-tumor effects. Zeolitic imidazolate framework-8 (ZIF-8) is considered as an attractive drug delivery vehicle for improved transport, release and anti-tumor effects of BIBR 1532. Herein, ZIF-8 and BIBR 1532@ZIF-8 were synthesized, respectively, and the physicochemical characterizations confirmed the successful encapsulation of BIBR 1532 in ZIF-8 coupled with an improved stability of BIBR 1532. ZIF-8 could alter the permeability of lysosomal membrane probably by the imidazole ring-dependent protonation. Moreover, ZIF-8 encapsulation facilitated the cellular uptake and release of BIBR 1532 with more accumulation in the nucleus. BIBR 1532 encapsulation with ZIF-8 triggered a more obvious growth inhibition of cancer cells as compared with free BIBR 1532. A more potent inhibition on hTERT mRNA expression, aggravated G0/G1 arrest accompanied with an increased cellular senescence were detected in BIBR 1532@ZIF-8-treated cancer cells. Our work has provided preliminary information on improving the transport, release and efficacy of water-insoluble small molecule drugs by using ZIF-8 as a delivery vehicle.

## 1. Introduction

Telomerase, a reverse transcriptase ribonucleoprotein complex, plays an important role in maintaining the telomere length in cells with high replicative potentials. Since its discovery in 1985 [1], telomerase has become an attractive target for new and more effective anticancer therapies due to its high associations with tumor development [2,3]. Cancer cells could acquire the self-proliferative capability via telomerase activation-dependent telomere length maintenance mechanisms. Considering that telomerase is constitutively overexpressed in the vast majority of human cancers [4,5,6,7], targeting the telomere/telomerase machinery offers a promising broad-spectrum anticancer therapeutic strategy by promoting telomere shortening, senescence and apoptosis. Human telomerase reverse transcriptase (hTERT) [8], the catalytic subunit of telomerase, plays a major role in telomerase activation in human cancers. Therefore, hTERT is considered a key target for telomerase inhibition-based cancer treatment [9].

Much work has been done on targeting telomerase to cure cancer, but unfortunately, no sufficiently potent and highly selective telomerase inhibitor has been available in clinic yet [10]. First, it takes a long time for telomerase inhibitors to shorten telomeres. Especially in tumors with long telomeres or long doubling times, the efficacy is always delayed [11,12,13]. Second, telomerase-mediated telomere elongation is not the only mechanism involved in tumorigenesis. In a small number of human tumors, telomerase shortening caused by long-term and low-dose treatment with telomerase inhibitors can be compensated by activation of alternative lengthening of telomere (ALT) [14,15,16]. It is clear that time and dose are important factors influencing the efficacy of telomerase inhibitors. BIBR 1532 is a well-known non-nucleosidic, non-peptidic telomerase inhibitor. It can bind to a conserved hydrophobic pocket (FVYL motif) of hTERT and inhibits enzymatic activity [17]. Accumulating evidence has shown the inhibition of BIBR 1532 on tumor cells via progressive telomere shortening and the consequent inhibition on cell proliferation [18,19,20]. However, BIBR 1532 is poorly soluble in water, which greatly reduces its uptake and utilization by cells. Moreover, growth inhibition is usually insignificant by short-term treatment with low concentration of BIBR 1532 [21], a common drawback of telomerase inhibitors.

Several studies have attempted to improve the bioeffects of BIBR 1532 by modifying its structure or combining it with other drugs or therapies. In one study, structural optimization based on BIBR 1532 was performed for an improved efficacy of telomerase inhibitors. A series of new compounds with different structural characteristics were designed, synthesized and screened. The compounds which used cyanide to replace methyl and simultaneously retained amide moieties were reported to have an improved telomerase inhibitory activity and moderate cytotoxicity as compared with BIBR 1532 [22]. Besides, several studies have adopted a combination of BIBR 1532 and other chemotherapeutics or therapies for a synergistic anti-tumor effect. For instance, the combination of BIBR 1532 and paclitaxel was reported to increase telomere dysfunction and inhibit cancer cells, enabling the observation of therapeutic outcome in a shorten time interval [23]. A similar study on the combinational application of BIBR 1532 and emodin demonstrated that, based on emodin-induced temporary telomere dysfunction by stabilizing the G4 structure at the protruding segment of the telomere end, the combined use of BIBR 1532 induced a permanent telomere dysfunction [24]. Another study found that low concentrations of BIBR 1532 were not toxic to lung cancer cells but could enhance ionizing radiation-induced apoptosis and cellular senescence [18]. However, little information is available concerning the drawbacks of low cellular uptake, inadequate delivery and limited bioeffects arising from the water insolubility of BIBR 1532 itself. Therefore, a proper drug delivery system is considered for improved transport, release and anti-tumor effects of BIBR 1532.

Metal-organic frameworks (MOFs) with three-dimensional pore structures are generally composed of metal ions (such as transition metals and lanthanide metals) and organic ligands (such as carboxylates, phosphonates, imidazole and phenols) [25]. To date, many attractive physicochemical characteristics as drug delivery tools have been identified for MOFs, including the ultra-high surface area, tunable porosity, diversified chemical functionalization, high loading capacity, good biocompatibility and the controllable release of the cargos [26,27,28,29,30].

Both traditional and emerging MOFs showed potentials for drug delivery and bio-orientation. The formation of a MOF layer by biomimetic mineralization on the surface of biological macromolecules can improve the stability of biological macromolecules, which is an important application of MOF in the field of biotechnology. Liang et al. demonstrated the ability to biomimetically mineralize several MOF materials, including ZIF-8, HKUST-1, Eu/Tb-BDC and MIL-88A. The results also showed that ZIF-8 has a good protective effect against the enzymatic activity of HRP, (PQQ)GDH and urease [31]. An emerging MOF, chiral MOF [32,33,34], can also be used as a drug carrier, enabling better biocompatibility, bioavailability and biodegradability of chiral drugs [35]. A study focused on and highlighted the necessity for MOF-based nanoplatforms in tumor treatment, summarizing the effects of MOF-based radiotherapy, chemotherapy, chemo-dynamic therapy, photothermal therapy, photodynamic therapy, starvation therapy and immunotherapy, and combination therapy against tumors [36]. MOF also have shown great potentials in clinical translation. In a study based on MOF combined with photothermal therapy and H_2_ therapy for the treatment of rheumatoid arthritis, Pt-MOF@Au@QDs/PDA had high biosafety and could maximize the treatment of rheumatoid arthritis, with great potential for clinical translation [37].

Zeolitic imidazolate framework-8 (ZIF-8), a member of the MOF family, consists of zinc metal properly coordinated with imidazole ligands. It exhibits as a cage-like three-dimensional network with a unique porosity similar to the zeolite. Thus, it possesses the dual ideal properties of zeolites and MOFs, such as tunable structure, monodisperse micropores, ultra-high surface area and structural flexibility [38]. More importantly, substantial data have demonstrated ZIF-8 as a promising drug delivery vehicle owning to its protonation effect induced by the contained imidazole ring. In the lysosomes, the imidazole ring is readily protonated in the acidic environment and causes the swelling of lysosomes, which increases the lysosomal membrane permeability and ultimately helps the lysosomal escape of the cargos. Actually, there have been many studies on the use of ZIF-8 as drug delivery vehicles. Scientists have encapsulated the genome editing machinery CRISPR/Cas9 with ZIF-8 and found that much more internalized CRISPR/Cas9 was released from lysosomes into cytoplasm and escaped from the lysosomal degradation when using ZIF-8 as a nanovehicle. This was found to be protonation-dependent lysosome escape [39]. In Hou’s work, encapsulating glucose oxidase with ZIF-8 could greatly improve the sensitivity and catalytic activity of glucose oxidase towards glucose with an enhanced stability [40]. Ding’s group encapsulated cytochrome C with ZIF-8 and showed that ZIF-8 encapsulation could protect cytochrome C from lysosomal proteolysis and ensure its safe delivery to cancer cells in an integral and active form [41]. Moreover, naked nucleic acids (DNA or RNA) are generally too large to penetrate into the cell membrane and are easily degraded by serum nucleases. In the study of Li’s group, plasmid DNA was successfully encapsulated in ZIF-8. ZIF-8 encapsulation greatly improved cellular uptake of plasmid DNA, increased the transfection efficiency and thus, achieved a more successful gene expression [42]. A recent study synthesized ZIF-8 on ZnO surface and the obtained ZnO@ZIF-8 structure enabled a high drug-loading efficacy of ciprofloxacin within the pore network of ZIF-8 and thus, achieved a controlled release and high antimicrobial efficiency of ciprofloxacin [43]. Ju et al. used ZIF-8 as a drug vector to deliver miR-491-5p into cancer cells and the results demonstrated that the ZIF-8 nanovector could stably release miR-491-5p for a long period and greatly improved the efficiency of miR-491-5p in inhibiting tumor growth [44]. All these reports have pointed out that ZIF-8 has great potential as a drug delivery vehicle.

Considering the water insolubility of the telomerase inhibitor BIBR 1532, the functional advantages of ZIF-8 as well as the lack of studies on addressing insignificant bioeffects of BIBR 1532, our study was the first to encapsulate BIBR 1532 with ZIF-8 and explore the effects of ZIF-8 as a drug delivery vehicle (Figure 1). Our present work demonstrated that the synthesized ZIF-8 facilitated the transport and release of BIBR 1532 within cells by altering the permeability of the lysosomal membrane. With ZIF-8 encapsulation, more BIBR 1532 accumulated in the nucleus and triggered a much more obvious growth inhibition of tumor cells as compared with free BIBR 1532.

## 2. Materials and Methods

### 2.1. Chemicals and Materials

Zinc nitrate hexahydrate (Zn(NO_3_)_2_ 6H_2_O) was purchased from Sigma-Aldrich. 2-methylimidazole was purchased from Macklin. Trypsin-EDTA was obtained from MACGENE. PBS buffer and DMEM (high glucose) were purchased from Hyclone. BIBR 1532 was obtained from Aladdin and the structure of BIBR 1532 was shown in Appendix A. Fluorescein, free acid (green fluorescence signal) was purchased from Coolaber. 6-aminofluorescein (6-AF, green fluorescence signal), N-hydroxysuccinimide (NHS) and 1-ethyl-3-(3-dimethylaminopropyl) carbodiimide (EDC) were obtained from Yuanye Biological. Hoechst 33,342 and Lyso-Tracker Red were purchased from Beyotime Biotechnology. Recombinant anti-lysosomal associated membrane protein 1 (LAMP1) antibody, recombinant anti-cathepsin B antibody and recombinant anti-GAPDH antibody were purchased from Abcam. Anti-rabbit IgG, HRP-conjugated antibody was purchased from Cell Signaling Technology. A549 cell line was obtained from American Type Culture Collection (ATCC).

### 2.2. Synthesis of Nanoparticles

ZIF-8 was synthesized according to a reported procedure with some modifications [45]. A measure of 0.2925 g of Zn(NO_3_)_2_ ·6H_2_O was dissolved in 3 mL of deionized water, and then 5.675 g of 2-methylimidazole was dissolved in another 20 mL of deionized water. The zinc nitrate solution was added to 2-methylimidazole solution with vigorous stirring. After stirring for 6 h, the product was collected and centrifuged at 15,000× *g* for 15 min. Finally, the as-synthesized nanoparticles were washed alternately with different ratios of anhydrous methanol/deionized water mixture (1:3 and 3:1) for 5 times. The product was then dried in a lyophilizer overnight to obtain ZIF-8 powder. All operations were performed at room temperature.

The loading of BIBR 1532 was achieved by synthesis of BIBR 1532@ZIF-8. Briefly, 0.011 g of Zn(NO_3_)_2_· 6H_2_O was dissolved in 0.113 mL of deionized water, then 0.86 mg of BIBR 1532 was dissolved in zinc solution; the mixed solution of zinc and BIBR 1532 was added to 0.752 mL of deionized water containing 0.213 g of 2-methylimidazole with vigorous stirring. After stirring for 6 h, the product was collected and washed. The product was then dried in a lyophilizer overnight to obtain the powder and stored at −80 °C. All operations were performed at room temperature.

To explore the transfer and localization of ZIF-8 in A549 cells, we labeled ZIF-8 with fluorescein (fluorescein@ZIF-8) according to a reported procedure with some modifications [46]. Briefly, 3.3 mg of fluorescein was dissolved in 0.3 mL of methanol and then mixed with 3 mL of deionized water containing 0.2925 g of Zn(NO_3_)_2_ ·6H_2_O. 5.675 g of 2-methylimidazole was dissolved in a further 20 mL of deionized water. The mixed solution of zinc nitrate and fluorescein was added to 2-methylimidazole solution with vigorous stirring. After stirring for 6 h, the product was collected and washed for several times. The product was then dried in a lyophilizer overnight to obtain fluorescein@ZIF-8 powder and stored at −20 °C. All operations were performed at room temperature.

The localization and accumulation of BIBR 1532 was observed by labeling BIBR 1532 with 6-AF (6-AF@BIBR 1532). BIBR 1532 was labeled with 6-AF according to a reported procedure with some modifications [47]. A 3.31 mg measure of BIBR 1532 was dissolved in 3 mL of absolute ethanol and 12 mL of deionized water, and then 5 mg of 6-AF was added while stirring. After stirring for 10 min, 10 mg EDS and 5.75 mg NHS were added to activate the carboxyl group of BIBR 1532. Stirring was continued for 6 h at room temperature. The reaction solution was then collected and washed several times. The product was then dried in a lyophilizer overnight to obtain the 6-AF@BIBR 1532 powder and stored at −80 °C. The whole experiment was performed in the dark.

The localization and accumulation of BIBR 1532@ZIF-8 was observed by synthesizing 6-AF@BIBR 1532@ZIF-8. A 0.011 g measure of Zn(NO_3_)_2·_ 6H_2_O was dissolved in 0.113 mL of deionized water, then 250 µL of 6-AF@BIBR 1532 obtained in the previous step was dissolved in the zinc solution. While stirring vigorously, the above mixed solution was added to 0.752 mL of deionized water containing 0.213 g of 2-methylimidazole. After stirring for 6 h, the product was collected, centrifuged at 15,000× *g* for 15 min and washed several times. The product was then dried in a lyophilizer overnight to obtain the powder and stored at −80 °C. The whole experiment was performed in the dark.

### 2.3. Characterization

The size and morphology of nanoparticles were observed by scanning electron microscope (SEM, s-4800, HITACHI, Tokyo, Japan) and transmission electron microscope (TEM, JEM2100Plus, JEOL, Tokyo, Japan). The size distribution and zeta potential of the nanoparticles were analyzed using dynamic light scattering (DLS, NanoBrook Omni, Holtsville, NY, USA). Powder X-ray diffraction (XRD) of the samples was recorded using a D8 Advance X-ray diffractometer (Bruker, Karlsruhe, Germany) at a voltage of 40 kV in the range of 2θ = 15–90° and a current of 40 mA. The formation of nanoparticles in the powder form was determined using a Fourier transform infrared (FT-IR) spectrometer (Thermo Scientific, Waltham, MA, USA) coupled with a Nicolet iN10 MX spectrograph. Thermal Gravimetric Analysis (TGA) was performed on TGA-Q50 (TA Instruments, New Castle, DE, USA). In total, 10 mg of samples were placed in an alumina crucible and heated in a continuous flow of nitrogen gas at a ramp rate of 10 °C/min from 0 °C up to 800 °C. The intracellular Zn content was determined using an inductively coupled plasma optical emission spectrometer (ICP-OES, Thermo Fisher Scientific, Pittsburg, PA, USA) to reflect the cellular uptake of ZIF-8.

### 2.4. BIBR 1532 Loading

6-AF@BIBR 1532 was dispersed in deionized water to get a stock concentration of 800 µg/mL and then diluted in a gradient to 400, 200, 100, 50, 25 and 12.5 µg/mL. The fluorescence intensity value of each well was measured using a microplate reader (Thermo Scientific, Waltham, MA, USA) with water as a blank control. The data were analyzed to obtain the standard curve equation: Y = 42066X + 17429, R^2^ = 0.9997 (Appendix A).

The fluorescence intensity value of the supernatant after centrifugation of 6-AF@BIBR 1532@ZIF-8 dispersion was measured and then substituted into the equation, giving a concentration of 0.269 mg/mL.

Drug loading was calculated using the following formula: Drug Loading = $\frac{0.6675\;-\;0.269\;\times \;(0.1128\;+\;0.752)}{7.333}$ = 6%. It means that each 100 mg of BIBR 1532@ZIF-8 contains 6 mg of BIBR 1532.

In this formula, 0.6675 mg was the weight of 6-AF@BIBR 1532 added for 6-AF@BIBR 1532@ZIF-8 synthesis. The concentration of 6-AF@BIBR 1532 in the supernatant after centrifugation of 6-AF@BIBR 1532@ZIF-8 dispersion was 0.269 mg/mL. The total volume of the synthetic 6-AF@BIBR 1532@ZIF-8 solvent was 0.8648 mL. The weight of the 6-AF@BIBR 1532@ZIF-8 product was 7.333 mg.

In the following study, we designed the incubation concentrations of BIBR 1532 and BIBR 1532@ZIF-8 at an equivalent concentration of BIBR 1532. We selected the concentrations of 10, 50, 100 µg/mL for BIBR 1532@ZIF-8, and the equivalent concentrations for BIBR 1532 were 0.6, 3, 6 µg/mL, respectively, with the corresponding ZIF-8 concentrations of 9.4, 47, 94 µg/mL.

### 2.5. Cell Viability Assay

A549 cells were maintained in Dulbecco’s modified Eagle’s Medium (DMEM) containing 10% fetal bovine serum, 100 U/mL penicillin and 100 µg/mL streptomycin at 37 °C in a humidified atmosphere of 5% CO_2_ and 95% air. The effects of ZIF-8, BIBR 1532 and BIBR 1532@ZIF-8 on cell viability were evaluated using CCK-8 assay (Dojindo Laboratories, Tokyo, Japan). Cells were seeded in 96-well plates (5 × 10^3^ cells/well) and cultured for 24 h. Cells were then treated with different concentrations of ZIF-8 (0, 50, 75, 100, 150 µg/mL) or different concentrations of ZIF-8 (9.4, 47, 94 µg/mL), BIBR 1532 (0.6, 3, 6 µg/mL) and BIBR 1532@ZIF-8 (10, 50, 100 µg/mL) for 24 h. Cell viability was assessed using standard procedures. Absorbance was measured at 450 nm using a microplate reader (Thermo Scientific, Waltham, MA, USA).

### 2.6. Inductively Coupled Plasma-Optical Emission Spectrometer (ICP-OES)

Cellular uptake of ZIF-8 was determined by measuring intracellular Zn content using ICP-OES. A549 cells were cultured on 35 mm petri dishes at a density of 2 × 10^5^ cells/dish. After adhesion, cells were incubated with 50 µg/mL ZIF-8 for different time intervals (1, 3, 6, 9, 12 and 24 h). After incubation, A549 cells were harvested and counted and then transferred to quartz beakers. A 3 mL measure of HNO_3_ was added to each quartz beaker. The quartz beaker was placed on a hot plate and kept at 150 °C for 2 h. Then, 1 mL of H_2_O_2_ was added for further heating digestion until digestion was complete. The solution volume was diluted to 3 mL with 2% HNO_3_. A series of Zn standard solutions (0.1, 0.5, 1, 5, 10, 50 and 100 ng/mL (ppb)) were prepared. Both standards and test samples were measured by ICP-OES. Cellular uptake of ZIF-8 was calculated from cell counts (µg/10^5^ cells).

### 2.7. Confocal Laser Scanning Microscopy

To explore the transfer and localization of ZIF-8, BIBR 1532 and BIBR 1532@ZIF-8, A549 cells were cultured on 35 mm dishes at a density of 4 × 10^5^ cells/dish. After attachment, the cells were treated with 100 µg/mL fluorescein@ZIF-8 or 6 µg/mL 6-AF@BIBR 1532 and 100 µg/mL 6-AF@BIBR 1532@ZIF-8 for 1 h in the dark. After incubation, the cells were incubated with fresh medium for a further 0, 2, 5 h. At each time point, the cells were fixed with 1mL of 2.5% glutaraldehyde for 15 min at room temperature. Afterwards, cells were stained with Lyso-Tracker Red and Hoechst 33342 at 37 °C for 20 min. The cells were then washed with PBS and loaded onto a laser scanning confocal microscope (Nikon A1, Tokyo, Japan) for observation.

To explore the nucleus accumulation of BIBR 1532 and BIBR 1532@ZIF-8, A549 cells were cultured on 35 mm dishes at a density of 4 × 10^5^ cells/dish and treated with 6 µg/mL 6-AF@BIBR 1532 or 100 µg/mL 6-AF@BIBR 1532@ZIF-8, respectively, for different time intervals. At 6, 12 and 24 h, cells were fixed with 4% paraformaldehyde for 15 min at room temperature followed by Hoechst 33342 staining at 37 °C for 10 min. Then cells were washed with PBS and loaded onto a laser scanning confocal microscope for observation.

### 2.8. Acridine Orange (AO) Staining

AO staining was used to reflect the change of lysosomal membrane permeability induced by ZIF-8. Cells were seeded onto 35 mm petri dishes at a density of 4 × 10^5^ cells/well. After attachment, cells were stained with AO at a final concentration of 10 µg/mL at 37 °C in the dark for 15 min. Afterwards, the cells were treated with or without 100 µg/mL ZIF-8 for different time periods (1, 3, 6, 12 and 24 h). At each time point, cells were fixed with 1 mL of 4% paraformaldehyde for 15 min at room temperature. The samples were then washed and loaded onto a fluorescence microscope for observation.

### 2.9. Western Blot Analysis

A549 cells were seeded on 6-well plates at a density of 9 × 10^5^ cells/well and cultured for 24 h. Then, cells were treated with different concentrations of ZIF-8 (0, 10, 25, 50, 75 and 100 µg/mL) for 24 h. At the end of the incubation period, the cells were lysed in RIPA lysis buffer (Beyotime Biotechnology, China) on ice for 15 min and protein was extracted. The lysate was centrifuged at 18,000× *g* for 5 min at 4 °C and the protein was collected. The protein concentrations of the samples were measured using a bicinchoninic acid assay (BCA) kit (Beyotime Biotechnology, China). Equal amounts (20 µg) of protein from the samples were separated by 12% sodium dodecyl sulfate-polyacrylamide gel electrophoresis. Proteins were transferred to 0.22 µm PVDF membranes (Millipore, Boston, MA, USA). Proteins were then blocked for 1 h in 5% BSA and incubated overnight at 4 °C with primary antibodies of lysosomal-associated membrane protein 1 (LAMP1) and cathepsin B, followed by 1 h of incubation with HRP-conjugated secondary antibody. Target protein bands were visualized using ECL-chemiluminescent kit (ECL-plus, Thermo Scientific, Waltham, MA, USA). Proteins levels were semi-quantitatively analyzed using ImageJ software. Glyceraldehyde-3-phosphate dehydrogenase (GAPDH) was used as a loading protein. The relative expression levels of LAMP1 and cathepsin B were normalized to the level of GAPDH.

### 2.10. Real-Time Quantitative Polymerase Chain Reaction (RT-qPCR) Analysis

hTERT mRNA level was detected using RT-qPCR analysis. Cells were seeded on 35 mm dishes at a density of 9 × 10^5^ cells/well and cultured for 24 h. Then, cells were treated with different concentrations of ZIF-8 (9.4, 47, 94 µg/mL), BIBR 1532 (0.6, 3, 6 µg/mL) and BIBR 1532@ZIF-8 (10, 50, 100 µg/mL) for 24 h (drug-loading efficiency is approximately 6%). After incubation, RNA was extracted from cells, followed by reverse transcription (FastFire Rapid Fluorescence PCR Premix (SYBR Green), TIANGEN, Beijing, China) and PCR amplification (Telomerase Activity Fluorescence Real-time Quantitative PCR Detection Kit (Human), KeyGEN BioTECH, Nanjing, China). Measurements were performed according to the manufacturer’s instructions. GAPDH was used as an internal control.

### 2.11. Cytokinesis-Block Micronucleus (CBMN) Assay

The micronucleus assay (MN) is one of the most popular methods to assess genotoxicity. The formation of micronucleus is easily recognizable in cytokinesis-blocked cells with their binucleate appearance [48]. Cell division block and telomere dysfunction induced by BIBR 1532 or BIBR 1532@ZIF-8 was explored using the cytokinesis-block micronucleus assay. In detail, cells were seeded in 35 mm dishes at a density of 4 × 10^5^ cells/well, then treated with 94 µg/mL ZIF-8, 6 µg/mL BIBR 1532 or 100 µg/mL BIBR 1532@ZIF-8. After incubation for 2 h, cytochalasin B was added with a final concentration of 5 µg/mL for one cell cycle (22 h for A549 cells) to ensure inhibition of cytoplasmic division to form binucleated cells, making it easier to observe micronucleus formation. Cytochalasin B was then removed and the cells were incubated with ZIF-8, BIBR 1532 or BIBR 1532@ZIF-8, respectively, for a further 48 h. At the end of the incubation period, the cells were fixed and then stained with Hoechst 33342 (100×) at 37 °C for 10 min. Then cells were washed and loaded onto a laser scanning confocal microscope for observation. Representative images were captured. The number of cells containing micronucleus and the total amount of micronucleus among the 300 cells in different groups were compared.

### 2.12. Cell Cycle Analysis

Cells were seeded in 6-well plates at a density of 4 ×10^5^ cells/well and cultured for 24 h. Then, the cells were treated with different concentrations of ZIF-8 (9.4, 47, 94 µg/mL), BIBR 1532 (0.6, 3, 6 µg/mL) or BIBR 1532@ZIF-8 (10, 50, 100 µg/mL) for 24 h. At the end of the treatment time, all the cells were collected and centrifuged at 1000× *g* for 5 min. Afterwards, cells were resuspended with 300 µL of pre-cooled PBS and then fixed with 700 µL of absolute ethanol at −20 °C overnight. Then cells were centrifuged, washed and stained with propidium iodide (PI) at 37 °C in the dark for 30 min. Cell cycle analysis was performed by flow cytometry (BD LSR-Fortessa, San Jose, CA, USA).

### 2.13. Senescence-Associated β-Galactosidase Detection

The effects of BIBR 1532 and BIBR 1532@ZIF-8 on cellular senescence were evaluated based on senescence-associated β-galactosidase (SA-β-Gal) staining using the Senescence Cells Staining Kit (Beyotime Biotechnology, Shanghai, China). Cells were seeded in 24-well plates at a density of 8 × 10^4^ cells/well and cultured for 24 h. Then, cells were treated with different concentrations of ZIF-8 (9.4, 47, 94 µg/mL), BIBR 1532 (0.6, 3, 6 µg/mL) or BIBR 1532@ZIF-8 (10, 50, 100 µg/mL) for 7 days. Cells treated with 100 nM DOX for 7 days were used as the positive controls. At the end of the treatment intervals, cells were subjected to SA-β-Gal staining according to the manufacturer’s instructions. For quantitative analysis of the β-galactosidase positive rate, 200 cells were counted in each group.

### 2.14. Statistical Analysis

Statistical analysis was performed using two-way ANOVA (GraphPad Prism 7.00) followed by Tukey’s multiple comparison test. All statistical parameters were presented as mean ± standard deviation (SD). *p* < 0.05 was considered significant.

## 3. Results

### 3.1. Physicochemical Characterizations of ZIF-8 and BIBR 1532@ZIF-8

The observation of ZIF-8 using scanning electron microscopy showed that ZIF-8 had a regular hexagonal or regular octagonal structure with a particle size of approximately 100–200 nm, which was further confirmed by the results from transmission electron microscopy (Figure 1A,B). The mean hydrated particle size of ZIF-8 was 129.98 ± 0.25 nm with a polydispersity index (PDI) of about 0.12 ± 0.01, indicating a relatively good dispersion. The zeta potential of ZIF-8 was tested as being 17.8 ± 1.3 mV (Figure 1E). The FT-IR results showed that most absorption peaks of the ZIF-8 sample came from the vibration of 2-methylimidazole: the band observed at 1589 cm^−1^ was assigned to the stretching vibration of the C=N group of 2-methyl imidazole. The bands in the range 1350–1500 cm^−1^ were assigned to the stretching vibration of the imidazole ring. Other bands in the 900–1350 cm^−1^ range were assigned to the stretching and bending modes of the imidazole ring (Figure 1H). The infrared spectroscopic results clearly demonstrated the successful synthesis of ZIF-8. The XRD patterns of ZIF-8 together with the corresponding simulated XRD patterns are shown in Figure 1I. The observed sharp and strong characteristic peaks were consistent with the simulated XRD patterns, suggesting that ZIF-8 had a rhombohedral dodecahedral structure. Few miscellaneous peaks were detected, indicating a relatively pure ZIF-8 crystal (Figure 1I). The mass change curve of ZIF-8 from TG analysis showed a long stable platform in the temperature range 25–400 °C, and the mass was decreased sharply as the temperature increased further, suggesting that the crystal skeleton of ZIF-8 began to decompose at higher temperatures (Figure 1J).

Then, ZIF-8 was loaded with BIBR 1532. The obtained BIBR 1532@ZIF-8 also maintained a regular hexagonal or octagonal shape as observed using SEM and TEM (Figure 1C,D). The mean hydrated particle size of BIBR 1532@ZIF-8 was 165.37 ± 0.48 nm, slightly larger than that of ZIF-8, with a polydispersity index (PDI) about 0.21 ± 0.01. The potential value of BIBR 1532 @ZIF-8 was -13.65 ± 0.93 (Figure 1F). BIBR 1532 was insoluble and poorly dispersed in water. It would precipitate rapidly within 1 day with a transparent supernatant appearing. In contrast, the dispersion of BIBR 1532@ZIF-8 in water remained relatively stable without precipitation for at least 1 week, suggesting an improved dispersion and stability of BIBR 1532 in water with ZIF-8 as a vector (Figure 1G). The FT-IR results of BIBR 1532@ZIF-8 showed that the bands observed at 1589 cm^−1^, 1350–1500 cm^−1^ and 900–1350 cm^−1^ were assigned to the vibration of 2-methylimidazole. In addition, the specific band of BIBR 1532@ZIF-8 was observed at 1300 cm^−1^, which corresponded to the stretching absorption of -OH in BIBR 1532. The results of FT-IR results suggested the successful synthesis of BIBR 1532@ZIF-8 (Figure 1H). The XRD results indicated that BIBR 1532@ZIF-8 had a rhombohedral dodecahedral structure similar to that of ZIF-8 (Figure 1I). The TGA curve of BIBR 1532@ZIF-8 showed that the weight decreased only by 10% even at temperature up to 750 °C (Figure 1J). Strikingly, the successful encapsulation of BIBR 1532 with ZIF-8 was again demonstrated by the changes in the curve.

### 3.2. ZIF-8 Localized in Lysosomes

First, the results from CCK-8 assay demonstrated that ZIF-8 did not induce obvious cytotoxicity in A549 cells in the concentration range of 0–100 µg/mL within the observed 24 h. However, when the concentration of ZIF-8 reached 150 µg/mL, a little cytotoxicity was exhibited (Figure 2A).

Then, the cellular uptake of ZIF-8 was examined within 24 h. A549 cells were exposed to 50 µg/mL ZIF-8 for 24 h and the cellular uptake of ZIF-8 was determined indirectly by the analysis of the intracellular zinc content using ICP-OES. The intracellular Zn concentration increased invariably with increasing exposure time of cells to ZIF-8. The result demonstrated that during the incubation period, cellular uptake of ZIF-8 was accelerated within 1 h, then gradually decreased between 1 and 12 h and reached saturation after 12 h (Figure 2B).

To study the transfer and localization of ZIF-8 in A549 cells, we labeled ZIF-8 with fluorescein (green). By comparing bright-field and fluorescence phase images, it confirmed that ZIF-8 was successfully labeled with fluorescein and moreover, the fluorescence signal could remain stable for at least 7 days (Figure 2C and Appendix A). Our experiment also explored the pH-responsiveness of ZIF-8 in vitro. ZIF-8 was dispersed in neutral (pH 7.4) and acidic (pH 5.0) solution, respectively, and its average particle size was monitored at 0, 30, 60, 120, 240 and 480 min. As shown in Figure 2D, the average particle size of ZIF-8 remained basically stable at neutral pH whereas it gradually decreased till undetectable at acidic pH, indicating that ZIF-8 was degraded in acidic solution.

A549 cells were exposed to 100 µg/mL fluorescein@ZIF-8 for 1 h. Fluorescein@ZIF-8 was then discarded and the cells were incubated with fresh medium. The transfer and localization of fluorescein@ZIF-8 was observed after addition of fluorescein@ZIF-8 for 1, 3 and 6 h. At 1 h, fluorescein@ZIF-8 was localized inside lysosomes, as demonstrated by the obvious yellow dots of lysosomes (Figure 2E). At 3 h, the lysosomes still showed a yellow fluorescence signal, but the green fluorescence signal in the cytoplasm decreased. At 6 h, the green fluorescence signal of ZIF-8 almost disappeared in both lysosomes and cytoplasm.

We further investigated the effect of ZIF-8 internalization on lysosomes. First, lysosomal membrane permeability was examined using AO staining. AO, a fluorescent dye that can enter lysosomes, is used to label lysosomes. It is present in intact lysosomes as an oligomer with red fluorescence and as a monomer in the cytoplasm with green fluorescence. As the lysosomal membrane stability decreases, AO is released from lysosomes into the cytoplasm, and an increase in the green fluorescence signal in the cytoplasm can be observed using laser scanning confocal microscopy. Therefore, the alteration on the ratio of green/red fluorescence intensity can reflect the change in lysosomal membrane permeability. In the present study, no obvious change on the ratio of green/red fluorescence intensity was observed in the control cells within 24 h. However, after 24 h of co-incubation with ZIF-8, the ratio of green/red fluorescence intensity in the cells increased, that is, the red fluorescence signal of AO in intact lysosomes gradually decreased and the green fluorescence signal of AO in the cytoplasm gradually increased, indicating that more AO dye was released from the lysosomes into the cytoplasm. The results suggested that ZIF-8 exposure induced a change in lysosomal membrane permeability (Figure 3A,B).

The expressions of cathepsin B (CTSB) and lysosomal associated membrane protein 1 (LAMP1) were detected to further explore the effect of ZIF-8 on lysosomal function. Cathepsin B (CTSB), an active hydrolase inside lysosomes, can mediate the degradation of extracellular particles caused by endocytosis. The expression of CTSB could reflect the degradative function of lysosomes [49]. Lysosomal-associated membrane protein 1 (LAMP1), the most abundant protein on the lysosomal membrane, plays an important role in lysosomes trafficking, maintenance of lysosomal acidification and homeostasis [50,51,52]. A549 cells were harvested after treatment with different concentrations of ZIF-8 (0, 10, 25, 50, 75 and 100 µg/mL) for 24 h. The results from western blot analysis showed that ZIF-8 treatment had no apparent effect on the expression of CTSB and LAMP1 in cells (Figure 3C).

### 3.3. BIBR 1532 Encapsulation by ZIF-8 Increased Delivery Efficacy

First, we labeled BIBR 1532 with 6-AF (green). The FT-IR results confirmed the successful labeling of BIBR 1532 by 6-AF (Figure 4A). Due to the coupling of the amino group of 6-AF with the carboxyl group of BIBR 1532, 6-AF@BIBR 1532 exhibited a weaker band at 3500–3350 cm^−1^ and 1700–1500 cm^−1^ associated with the stretching vibration of the amino group of 6-AF. Compared with BIBR 1532, 6-AF@BIBR 1532 exhibited a weaker band corresponding to the stretching of -COOH at 1300 cm^−1^, which fully demonstrated the successful labeling of BIBR 1532 with 6-AF.

Then, we encapsulated 6-AF@BIBR 1532 with ZIF-8 and confirmed the successful synthesis of 6-AF@BIBR 1532@ZIF-8 by FT-IR results (Figure 4B). The infrared spectrum of 6-AF@BIBR 1532@ZIF-8 showed that the band observed at 1589 cm^−1^ was assigned to the stretching vibration of the C=N group of 2-methylimidazole; the bands in the range 1350–1500 cm^−1^ were assigned to the stretching vibration of the imidazole ring; the bands in the range 900–1350 cm^−1^ were assigned to the stretching and bending modes of the imidazole ring; the band at 1300 cm^−1^ corresponded to the stretching of -COOH. Due to the carboxyl-amino coupling, 6-AF@BIBR 1532@ZIF-8 exhibited weaker stretching bands of -COOH at 1300 cm^−1^, indicating the successful synthesis of 6-AF@BIBR 1532@ZIF-8. Furthermore, by comparing the images of the bright-field and fluorescence images, we demonstrated that the encapsulation of ZIF-8 did not affect the detection of the fluorescence signals (Figure 4C) and the fluorescence signal remained stable for at least one week (Appendix A).

Figure 4D illustrated the different transfer pathways and localization of 6-AF@BIBR 1532 and 6-AF@BIBR 1532@ZIF-8 in A549 cells. At 1 h after the addition of 6-AF@BIBR 1532 or 6-AF@BIBR 1532@ZIF-8, the green fluorescein was localized inside lysosomes, as shown by the obvious yellow dots in lysosomes (Figure 2D). After 3 h of incubation, green fluorescence of 6-AF@BIBR 1532 and 6-AF@BIBR 1532@ZIF-8 in lysosomes decreased. In 6-AF@BIBR 1532@ZIF-8-incubated cells, the fluorescence was distributed in the cytoplasm and even a very small amount of fluorescence signal appeared in the nuclear region. In 6-AF@BIBR 1532-incubated cells, the decreased fluorescence in lysosomes was not accompanied by an increase in the cytoplasm or nucleus. After 6 h of incubation, only a very small amount of fluorescence was detected in the nuclear region of 6-AF@BIBR 1532-treated cells. Comparatively, more fluorescence was concentrated in the nucleus in 6-AF@BIBR 1532@ZIF-8-treated cells. The results showed that encapsulation of BIBR 1532 in ZIF-8 led to an increased release of BIBR 1532 into the cytoplasm and the sequent transfer into the nucleus.

We also evaluated the accumulation of 6-AF@BIBR 1532 and 6-AF@BIBR 1532@ZIF-8 in the nucleus within 24 h. Free BIBR 1532 accumulated slowly in the nucleus. The fluorescence intensity in the nucleus increased slowly from 6 to 24 h, suggesting a relatively limited cumulative amount of BIBR 1532 within 24 h. With ZIF-8 encapsulation, more BIBR 1532 accumulated in the nucleus. The fluorescence intensity in the nucleus increased sharply from 6 h to 24 h, suggesting a steady accumulation of BIBR 1532 in the nuclear region (Figure 4E,F). Herein, the encapsulation of BIBR 1532 in ZIF-8 successfully increased the nuclear accumulation of BIBR 1532.

### 3.4. ZIF-8 Encapsulation Enhanced the Treatment Efficacy of BIBR 1532

The advantage of encapsulating BIBR 1532 with ZIF-8 was evaluated from several aspects. As demonstrated in Figure 5A, the treatment with 0.6 µg/mL BIBR 1532 did not cause a decrease in cell viability, whereas the treatment with the equivalent BIBR 1532 encapsulated by ZIF-8 (100 µg/mL BIBR 1532@ZIF-8) caused an 18% decrease in cell viability. This result suggested a remarkable increase in the cytotoxicity of BIBR 1532 with ZIF-8 as a delivery vector. This kind of enhanced cytotoxicity was more evident in the higher concentration group. Only a 20% decrease of cell viability was induced by the treatment of 6 µg/mL BIBR 1532 within 24 h whereas the equivalent BIBR 1532 contained in 100 µg/mL BIBR 1532@ZIF-8 caused an 80% decrease of cell viability. Therefore, it demonstrated that the cytotoxicity of BIBR 1532 could be enhanced significantly by ZIF-8 encapsulation.

To explore the inhibitory effects of BIBR 1532 and BIBR 1532@ZIF-8 on telomerase activity, hTERT mRNA levels were detected after cells were incubated with different concentrations of BIBR 1532 or BIBR 1532@ZIF-8. The results showed that the mRNA levels of hTERT in A549 cells treated with 0.6, 3 and 6 µg/mL BIBR 1532 were decreased to ~83%, ~72% and ~61%, respectively. A dose-dependent inhibition of hTERT mRNA levels was induced by BIBR 1532 alone. An equivalent amount of BIBR 1532 encapsulated by ZIF-8 induced a much more pronounced decrease in hTERT mRNA levels. It showed that when A549 cells were treated with 10, 50 and 100 µg/mL BIBR 1532@ZIF-8 for 24 h, the hTERT mRNA levels were decreased to ~21%, ~20% and ~17%, respectively. Moreover, no decreases in hTERT mRNA levels were caused by ZIF-8 alone (Figure 5B). Thus, ZIF-8 encapsulation could significantly enhance the inhibitory effect of BIBR 1532 on hTERT mRNA levels.

Micronucleus (MN) is the small circular or elliptical nucleus located in the cytoplasm, completely separated from the nucleus. When cells were exposed to genotoxic insults, the chromosomes in the cells are usually lost or broken and fail to enter the daughter cell with mitosis. The end-to-end fusion of the broken chromosomes occurs, forming one or more small nucleus in the cytoplasm [53,54]. The appearance of micronucleus may reflect telomere dysfunction [55,56]. In the present study, cells were treated with 94 µg/mL ZIF-8, 6 µg/mL BIBR 1532 and 100 µg/mL BIBR 1532@ZIF-8 for 72 h, and then the formation of the micronucleus was observed using CLSM. As a result, ZIF-8 treatment alone did not promote the formation of the micronucleus in cells. A micronucleus was observed in BIBR 1532-treated cells and a much more obvious micronucleus formation was exhibited in BIBR 1532@ZIF-8-treated cells (Figure 5C). Then, a statistic was made on the micronucleus formation of 300 cells. As a result, neither the number of cells containing a micronucleus nor the total amount of micronuclei among the 300 cells showed any difference between the control group and the ZIF-8-treated group, suggesting that ZIF-8 treatment alone had no impact on micronucleus formation. As expected, treatment with free BIBR 1532 induced an obvious micronucleus formation, but most binuclear cells contained only one micronucleus, whereas BIBR 1532@ZIF-8 led to the formation of a micronucleus in more cells and multiple micronucleus in one cell. According to the statistic, treatment with 100 µg/mL BIBR 1532@ZIF-8 induced an 8.3% increase in the percentage of micronucleus-containing cells and a 14.3% increase in the percentage of the micronucleus as compared with the group treated with the equivalent free BIBR 1532 (6 µg/mL). Taken together, the encapsulation of BIBR 1532 in ZIF-8 promoted the formation of the micronucleus, indicating an increased telomere dysfunction.

Next, the effects of BIBR 1532 and BIBR 1532@ZIF-8 on cell cycle distribution were explored. Compared with that of the control group, no alteration on cell cycle distribution was detected in ZIF-8-treated cells. A slight G0/G1 arrest was induced by the treatment of free BIBR 1532 relative to that of the control group. The percentages of cells at the G0/G1 phase upon the treatment of 0.6, 3 and 6 µg/mL BIBR 1532 were 45.82%, 45.36% and 54.50%, respectively. Notably, the percentages of cells at the G0/G1 phase after treatment with 10, 50 and 100 µg/mL BIBR 1532@ZIF-8 were increased to 47.00%, 80.54% and 82.88%, respectively, indicating that the encapsulation of BIBR 1532 with ZIF-8 aggravated BIBR 1532-induced G0/G1 phase arrest (Figure 6A,B).

The effect of BIBR 1532 with or without ZIF-8 encapsulation on cellular senescence was assessed using senescence-associated β-galactosidase (SA-β-Gal) staining. As demonstrated in Figure 7A, an obvious cellular senescence was exhibited in doxorubicin (DOX)-treated cells, which was used a positive control. Meanwhile, no obvious cellular senescence occurred in either the control group or ZIF-8-treated group. Free BIBR 1532 treatment induced a dose-dependent cellular senescence as evidenced by the increased number of green-stained cells with the increasing BIBR 1532 concentrations. The percentages of SA-β-Gal positive cells upon the treatment of 0.6, 3 and 6 µg/mL BIBR 1532 were 1.8%, 2.8% and 4.5%, respectively (Figure 7B), whereas in the BIBR 1532@ZIF-8-treated group, cell density obviously decreased with increasing exposure concentrations. Notably, the percentages of SA-β-Gal positive cells after treatment with 10, 50 and 100 µg/mL BIBR 1532@ZIF-8 were increased to 3.2%, 30.7% and 89.0%, respectively. The inhibition of cell proliferation induced by BIBR 1532@ZIF-8 was again confirmed here. Therefore, the encapsulation of BIBR 1532 in ZIF-8 aggravated BIBR 1532-induced cellular senescence.

## 4. Discussion

It has been widely accepted that poor solubility poses great limitations for the bioavailability and efficiency of many drugs. Fortunately, the rapid nanotechnology has provided powerful solutions to this problem owing to the great advantages of nanocarriers. For instance, polymer nanocomposites were used to improve the solubility of curcumin for an enhanced antioxidant and anti-inflammatory effect in vitro [57]. The loading of water-insoluble tripterin with bilayer structured liposomes has achieved a better solubility, bioavailability and lower toxicity [58]. Other studies have also achieved more significant therapeutic effects by improving drug solubility [59,60]. In our present study, encapsulating BIBR 1532 with ZIF-8 was found to increase the stability of BIBR 1532 in water as well as its cellular uptake.

ZIF-8 is pH sensitive because the cleavage of the ligand coordination bond between the zinc ion and 2-methylimidazole in ZIF-8 readily occurs in acidic environments. Thus, ZIF-8 is considered to a promising drug vehicle for pH-sensitive release of the loaded drugs [61]. Our experiment also confirmed its pH-responsiveness. As shown in Figure 2D, the average particle size of ZIF-8 remained unaltered when dispersed in the neutral environment with pH 7.4, whereas in the acidic environment with pH 5.0, its average particle size decreased significantly till undetectable within the tested 480 min. Combined with our experimental results in Figure 2E and the pH-responsiveness capability of ZIF-8, it figured out that ZIF-8 was captured by the lysosome after entering the cell as shown by the colocalization of red fluorescence (Lyso-Tracker) and green fluorescence (fluorescein-labeled ZIF-8) at 1 h. As the incubation time increased, the green fluorescence intensity in the cytoplasm gradually decreased, indicating that ZIF-8 in the cytoplasm was continuously captured by lysosomes. After entering the acidic environment of the lysosomes, the coordination bond between zinc and imidazole was broken. After 6 h, no green fluorescence was observed in the cytoplasm or lysosomes, suggesting that most of the ZIF-8 and fluorescein had been cleared by the cells.

The loading efficiency of ZIF-8 largely depends on the loaded content. In a study where curcumin was encapsulated by ZIF-8, the drug loading capacity was up to 10.89% [62]. Another report showed that ZIF-8 encapsulated the chemotherapy drug rapamycin with a high drug loading rate of 9.39% [63]. However, the loading rate of ZIF-8 for the gene editing platform CRISPR/Cas9 was only 1.2% [39], and the loading rate of plasmid DNA was only 2.5% [42]. The drug loading rate of ZIF-8 for BIBR 1532 in our present study was 6%. The loading capability of ZIF-8 towards BIBR 1532 was not very striking from the above data. However, after being encapsulated into ZIF-8, BIBR 1532 actually showed much more significant inhibitory effects towards cancer cells. This suggested the feasibility of using ZIF-8 as a drug delivery vehicle for an improved BIBR 1532 efficacy in spite of a relatively general loading rate.

The proton sponge effect is acknowledged as one of the important mechanisms of lysosomal escape. Histidine-rich molecules exhibit a buffering effect upon protonation of the imidazole ring. The protonation induces the flux of ions and water into the lysosomal environment, increasing the lysosomal membrane permeability. The lysosomal membrane then ruptures and releases the trapped components [64,65]. In addition to the proton sponge effect, other mechanisms of lysosomal escape have also been reported including the formation of pores in the lysosomal membrane [66], lysosomal membrane fusion [67] and the photochemical disruption of lysosomal membrane [68]. Our work has suggested that ZIF-8-facilitated lysosomal escape predominately depended on the proton sponge effect of 2-methylimidazole. Actually, the proton sponge effect has also been exploited by other scientists and many novel materials have been developed with the aim of promoting lysosomal escape of functional molecules and reducing the unnecessary lysosomal degradation. For instance, Chen’s group reported the use of metal (Fe^III^ or Al^III^)-phenolic (polyphenol tannic acid) networks as a coating for nanoparticles, which could achieve lysosomal escape of nanoparticles through the proton sponge effect [69]. Xi et al. also designed the functional nanoparticles (PG-FAPEP) consisting of insulin with dual surface decorations of folate and charge-convertible tripeptide. Such functional nanoparticles could also trigger the proton sponge effect in acidic lysosomes and help the lysosomal escape of insulin [70]. Another report demonstrated that by immobilizing the highly branched and rigid cationic polymers on silica nanoparticles, the proton sponge effect with osmotic swelling and lysosome rupture could be achieved [71].

Nanoparticles typically enter cells via an energy-dependent process known as endocytosis. During endocytosis, nanomaterials are usually initially confined to vesicular structures such as endosomes, lysosomes, phagosomes or macropinosomes [72]. Different endocytic pathways will lead to different fates after nanomaterials enter the cell [73]. To date, endocytic pathways can be classified and summarized into the following 7 types: macro pinocytosis, phagocytosis, clathrin independent/dynamin-dependent endocytosis, clathrin-mediated endocytosis, caveolae-mediated endocytosis, clathrin independent/dynamin-independent endocytosis and fusion [74,75]. Different internalization pathways are mediated and regulated by different lipids and transporters. Our experiments showed that ZIF-8 and BIBR 1532@ZIF-8 were first captured by lysosomes after entering the cell, suggesting that ZIF-8 and BIBR 1532@ZIF-8 entered the cell by endocytosis. However, the actual endocytosis pathway needs to be further investigated in the following work.

In our work, the concentrations of BIBR 1532 used were 0.6, 3 and 6 µg/mL, corresponding to the molar concentrations of 1.81, 9.05 and 18.11 µM, respectively. No obvious G0/G1 phase arrest was induced in cells treated with 1.81 or 9.05 µM BIBR 1532 within 24 h, while cells treated with 18.11 µM BIBR 1532 showed a slight cell cycle arrest. Similarly, the number of senescent cells increased with the increasing concentrations of BIBR 1532. With ZIF-8 as a drug carrier, more BIBR 1532 accumulated in the nucleus and a remarkable cell cycle arrest was observed at 24 h especially in 50, 100 µg/mL BIBR 1532@ZIF-8-treated cells. No cell apoptosis was detectable (data not shown). With the incubation of BIBR 1532@ZIF-8 for 7 days, a remarkable decrease in cell number accompanied with increased cellular senescence was observed. Cellular senescence is described as a state of permanent cell cycle arrest in response to different damaging stimuli [76]. In the present study, it was reasonable that BIBR 1532@ZIF-8 induced cell cycle arrest followed by cellular senescence. Besides destroying tumor cells directly, pro-cellular senescence is an alternative to limit tumor cells. The relatively high concentrations of BIBR 1532 have been reported to induce apoptosis or destroy tumor cells directly. A study showed that LN18 cells treated with 25 µM BIBR 1532 showed flat and enlarged cell morphology. When the concentration of BIBR 1532 was increased to 100 µM, cell morphology was narrowed and the nuclear membrane was lost [77]. It figured out that the encapsulation of BIBR 1532 in ZIF-8 not only decreased the dosage of BIBR 1532 needed for a comparable inhibition effect towards tumor cells but also acts in a milder manner of promoting cell cycle arrest and senescence of tumor cells rather than destroying tumor cells directly.

Our work was the first to encapsulate hydrophobic BIBR 1532 with ZIF-8 for an improved BIBR 1532 efficacy. Besides of our study, only one study was available with the involvement of BIBR 1532 loading. In this study, peptide dendrimeric prodrug nanoassembly (PDPN) acted as a nanocarrier to load BIBR 1532 and DOX, thus constructing a telomerase-terminated nanoplatform to reverse DOX resistance [78]. As illustrated in Appendix A, the similarities and differences were analyzed between this study and our work. The similarities between the two papers reflected the common theoretical basis which clarified the practical problems in the application of BIBR 1532 and the necessity of loading BIBR 1532 with nanocarriers. The differences between the two papers were highlighted in the aspect of key observations, drug release, the bio-effects of BIBR 1532 and the significance of the work. In our present study, after BIBR 1532-bearing ZIF-8 was endocytosed into cells and localized in lysosome, ZIF-8 was degraded in the acidic environment and BIBR 1532 was released, causing the proliferation inhibition, cell cycle arrest and cellular senescence of tumor cells; whereas in this reported study [78], BIBR 1532 was released from the PEG layer and then weakened mitochondria protection, enhanced ROS production and meanwhile, promoted the apoptosis of DOX-resistant tumor cells. In any case, both studies addressed the essentials of proper drug delivery vehicles for an improved efficacy of hydrophobic BIBR 1532.

## 5. Conclusions

In the present study, ZIF-8 and BIBR 1532@ZIF-8 were synthesized, respectively, and the physicochemical characterizations confirmed the successful encapsulation of BIBR 1532 by ZIF-8 coupled with an improved stability of BIBR 1532. ZIF-8 facilitated the transport and release of BIBR 1532 within cells via the increased permeability of lysosomal membrane. As compared with free BIBR 1532, ZIF-8 encapsulation facilitated more of an accumulation of BIBR 1532 in the nucleus, leading to a more obvious growth inhibition of cancer cells. A more potent inhibition on hTERT mRNA expression, an aggravated G0/G1 arrest accompanied with an increased cellular senescence was detected in BIBR 1532@ZIF-8-treated cancer cells. Our work has provided preliminary information on improving the transport, release and efficacy of water-insoluble small molecule drugs by using ZIF-8 as a drug delivery vehicle.

## Data Availability

Not applicable.

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
