# Peer review of "Zeolitic Imidazolate Framework-8 (ZIF-8) as a Drug Delivery Vehicle for the Transport and Release of Telomerase Inhibitor BIBR 1532"

_nanomaterials, 2023, doi:10.3390/nano13111779_

Round 1
Reviewer 1 Report
This manuscript describes the usage of ZIF-8 as a drug delivery system for BIBR 1532. The work provides information concerning the improvement of transport and release of water-insuluble drug using ZIF-8 drug carrier. The manuscript is comprehensive and very sophisticated. The Authors used a wide range of techniques for characterization of synthesized carrier (e.g. SEM, TEM, XRD, FT-IR, TGA, ICP-OES). Moreover, many advanced biological tests were employed for characterization of drug-carrier system (e.g. cell viability, confocal laser microscopy, Western blot analysis, RT-qPCR, CBMN assay). Before the acceptance of the article in Nanomaterials the following issues should be addressed:
1. Instead of unit "rpm" in section 2.2 please provide transformation into "g" unit which is more representative.
2. In Fig. 1 (B, D) please provide more visible scale.
3. Less than half of the references was published in last five years ( 31/65). Please, add some more new references.
Author Response
- In section 2.2, we converted the unit of the centrifugation from speed to centrifugal force.
- We have added a new scale as demonstrated in Fig. 1 (B, D).
- We have enriched the introduction and discussion sections, and also updated the references.
Reviewer 2 Report
The work describes the synthesis and the physicochemical characterizations of ZIF-8 and BIBR 1532 encapsulating ZIF-8 and the ability of the latter to increase the permeability of the lysosomal membrane and by that to facilitate the transport and release of BIBR 1532 within cells. An increase of the growth inhibition of cancer cells is shown upon encapsulation of BIBR 1532 in ZIF-8 which is important for the improvement of cancer treatments. As a more general result the work demonstrates the ability of ZIF-8 as a drug delivery vehicle to improve the transport, release and efficacy of water-insoluble drugs. The work is very interesting and publication is recommended in Nanomaterials after some minor revisions reported below.
A zeta potential of ZIF-8 of 17.78±1.35 mV is reported in the manuscript which should be justified, why are the particles positive?
In addition, it is meaningless to report two decimal figures if the error is larger than one, 17.8±1.3 mV or 18±2 mV is suggested.
The scale bar of TEM images is not reported in figure 1B,D.
The scale bar of SEM is not clear in figure 1A,C.
The introduction should be enlarged, mentioning emergent classes of MOFs which could have relevant potentiality for drug deliveries and bio-oriented applications. For examples, MOFs inspired by biomineralization processes ( Liang, K., et al. . Nat Commun 6, 7240 (2015); di Gregorio, M.C., et al. Nat Commun 12, 957 (2021); Nat Commun 2020, 11, 380) exhibited unique fundamental and applicative properties, translating into laboratory materials synthetic and functional aspects that are proper of living organisms.
Another important feature for the interaction with drugs and biomolecules is the possibility to have chiral vectors. Recent literature both reviewed this aspect (Gong et al, Chem. Rev. 2022, 122, 9, 9078–9144; Ma et al Nanoscale, 2022,14, 13405-13427; Israel Journal of Chemistry, 2021, 61, 708-726) and highlighted paradoxical and multifunctional chiral metal-organic frameworks (J. Am. Chem. Soc. 2022, 144, 50, 22838–22843; Angew. Chem. Int. Ed., 2022, e202205238; Chirality, 2022, 34, 1419-1436;)
Author Response
- We have synthesized ZIF-8 using only 2-methylimidazole and Zn(NO3)2 •6H2O in aqueous solution without any other modifications. We have re-determined the positive potential properties of ZIF-8 and believe that this result is due to the presence of zinc ions. In addition, we have cited several publications that confirm the positive potential nature of ZIF-8 (Qiu, J., et al. Journal of colloid and interface science 642, 810-819 (2023); Alsaiari, S. K., et al. Journal of the American Chemical Society 140, 143-146 (2018); Babayevska, N., et al. Biomaterials advances 144, 213206 (2023); Peng, S., et al. ACS applied materials & interfaces 11, 35604-35612 (2019); Wu, S., et al. Advanced science (Weinheim, Baden-Wurttemberg, Germany) 9, e2103534 (2022); Peng, L., et al. Drug delivery 29, 1075-1085 (2022); Ding, L., et al. ACS applied materials & interfaces 12, 36906-36916 (2020)).
- We have modified 17.8±1.35 mV to 17.8±1.3 mV as your suggestion.
- We have re-added the scale bars in Figure 1A-D.
- Following your helpful suggestions, we have added new references in the introduction section to illustrate the potential of traditional MOFs and emerging MOFs (formed by biomineralization or chiral MOFs) for drug delivery and the promising clinical translation.
Reviewer 3 Report
It is an interesting article. However, the follows must be addressed.
1.Discuss Loading capacity of BIBR 1532 in ZIF 8.
2.Quantify the cytotoxicity or cell density of BIBR 1532 in Figure 7 and explain the green spots (senescence) among cells in caption.
3.Discuss endocytosis pathway of BIBR 1532/ZIF8 in lung cancer cells
4.Summarize the achivement or performance using a table by comparing with other publishing.
5.Add the structure of BIBR 1532 in supplementary materials
Author Response
- The loading capacity of BIBR 1532 in ZIF-8 has been discussed in the Discussion section.
- We have quantified the experimental results of cellular senescence and added new information to the Methods and Results sections.
- We have listed and summarized several pathways of endocytosis and have indirectly shown that BIBR 1532@ZIF-8 entered the cell by endocytosis, but it is still uncertain by which pathway BIBR 1532@ZIF-8 entered the cell. We plan to explore it further in the following work.
- Thanks for your advice. Another publication concerning the loading of BIBR1532 (Reference 78#) was found and we comprehensively analyzed its similarities and differences from our present study as summarized in Table 1 in Discussion section.
- We have already included the structure of BIBR 1532 in the supplementary materials.